# Automated Movement Analysis to Predict Cerebral Palsy in Very Preterm Infants: An Ambispective Cohort Study

**DOI:** 10.3390/children9060843

**Published:** 2022-06-07

**Authors:** Kamini Raghuram, Silvia Orlandi, Paige Church, Maureen Luther, Alex Kiss, Vibhuti Shah

**Affiliations:** 1Division of Neonatology, Department of Paediatrics, University of Toronto, Toronto, ON M5G 1X8, Canada; kamini.raghuram@sinaihealth.ca (K.R.); paige.church@sunnybrook.ca (P.C.); 2Bloorview Research Institute, Holland Bloorview Kids Rehabilitation Hospital, Toronto, ON M4G 1R8, Canada; silvia.orlandi9@unibo.it; 3Department of Electrical, Electronic, and Information Engineering, Guglielmo Marconi (DEI), University of Bologna, 40136 Bologna, Italy; 4Neonatology and Neonatal Follow-Up, Women and Babies Program, Sunnybrook Health Sciences Centre, Toronto, ON M4N 3M5, Canada; maureen.luther@sunnybrook.ca; 5Institute for Clinical Evaluative Sciences (ICES), Toronto, ON M4N 3M5, Canada; alex.kiss@ices.on.ca; 6Department of Paediatrics, Mount Sinai Hospital, Toronto, ON M5G 1X5, Canada; 7Institute of Health Policy, Management and Evaluation, University of Toronto, Toronto, ON M5T 3M6, Canada

**Keywords:** preterm infant, cerebral palsy, general movements assessment, early detection, neonatal follow-up, early intervention, machine learning, computer vision

## Abstract

The General Movements Assessment requires extensive training. As an alternative, a novel automated movement analysis was developed and validated in preterm infants. Infants < 31 weeks’ gestational age or birthweight ≤ 1500 g evaluated at 3–5 months using the general movements assessment were included in this ambispective cohort study. The C-statistic, sensitivity, specificity, positive predictive value, and negative predictive value were calculated for a predictive model. A total of 252 participants were included. The median gestational age and birthweight were 27^4/7^ weeks (range 25^6/7^–29^2/7^ weeks) and 960 g (range 769–1215 g), respectively. There were 29 cases of cerebral palsy (11.5%) at 18–24 months, the majority of which (*n* = 22) were from the retrospective cohort. Mean velocity in the vertical direction, median, standard deviation, and minimum quantity of motion constituted the multivariable model used to predict cerebral palsy. Sensitivity, specificity, positive, and negative predictive values were 55%, 80%, 26%, and 93%, respectively. C-statistic indicated good fit (C = 0.74). A cluster of four variables describing quantity of motion and variability of motion was able to predict cerebral palsy with high specificity and negative predictive value. This technology may be useful for screening purposes in very preterm infants; although, the technology likely requires further validation in preterm and high-risk term populations.

## 1. Introduction

Early detection of cerebral palsy (CP) in high-risk infants using the General Movements Assessment (GMA) is now considered one of the recommended assessment tools in neonatal follow-up [1]. Certainly, this has included cohorts of late preterm infants with and without neonatal encephalopathy [2,3], and a heterogeneous population of preterm infants with a median gestational age of 30 weeks [4]. It is important to identify and intervene early to potentially improve function and outcomes while neuroplasticity is at its peak [5]. A study evaluating targeted early intervention programs demonstrated clear efficacy when children were recruited based on GMA [6].

However, limited access to clinical expertise in GMA prevents its universal use [7]. In order to address this limitation, we embarked on a project to automate the GMA [8]. Similarly, several other groups [9,10,11,12] have also attempted to automate the GMA. One review summarized such technologies [13] and a large systematic review found them to have a pooled sensitivity and specificity of 73% and 70%, respectively [14]. However, with the exception of one other group [15], these studies have used convenience samples rather than consecutively recruited patients and have small sample sizes with which they have validated their technology. For example, one study’s discrimination of abnormal movements in infants with abnormal GMA clinically was lower because of the limited number of cases available [16]. Several technologies have relied on direct sensing (wearable techniques), which can be cumbersome and affect the infant’s movement [12,17,18,19,20,21]. Further, these studies have used a mixed population of infants with a range of possible outcomes, and thus, potentially with different movement patterns in the fidgety period [9,22,23]. These factors ultimately limit the technology’s validity to date and the ability to translate the technology into clinical practice.

The aim of this study was to apply a novel 2D video-based analysis method to a set of videos of preterm infants < 31 weeks GA at a tertiary-care neonatal follow-up clinic and to develop a statistical model to predict CP. We hypothesize that similar to other studies [9,15], the quantity of motion and the variability of motion will provide a strong predictive model for cerebral palsy, but may not reach the sensitivity of clinical general movements because of the inability to capture movement in three dimensions and the inability to characterize distal limb movements accurately. We anticipate that this simple technology may function as an initial screen for preterm infants, make the GMA more accessible and, ultimately, connect infants in a timely fashion with early intervention programs.

## 2. Materials and Methods

### 2.1. Study Design

An ambispective cohort study design was used for this collaborative clinical and engineering project undertaken at Sunnybrook Health Sciences Centre (SHSC) and the Bloorview Research Institute (BRI). This study design was chosen to adequately power the predictive model developed in this study. Ethics approval was obtained from the Research Ethics Boards at both centers. (Sunnybrook Health Sciences Centre (protocol code 174-2017, date of approval: 20 June 2017), Bloorview Research Institute (protocol code 18-768 and date of approval: 13 April 2018), University of Toronto (protocol code 38697, date of approval: 13 January 2020)). Informed parental consent was obtained at the time of the first clinical visit and confirmed at all subsequent visits for the prospective part of the study; for the retrospective study, consent was waived as a result of the study design. The STROBE checklist was used for reporting [24].

### 2.2. Setting

For the prospective portion of the study, patients were recruited from a single tertiary care neonatal follow-up clinic at SHSC between July 2017 and August 2018. Caregivers who opted not to be involved in the study were excluded. Enrolled patients were assessed between 3–5 months CA and 18–24 months as per the schedule of visits for the follow-up clinic. The video for automated analysis was acquired between 3 and 5 months CA, while the detailed developmental assessment was completed between 18 and 24 months CA. All patients were followed in clinic through December 2019, at which point the study was concluded. Data were collected on an ongoing basis throughout the study period. For the retrospective portion of the study, a previously-described video database of infants recorded at 3–5 months CA between 2009 and 2015 was used [8].

### 2.3. Participants

Preterm infants with GA < 31 weeks or birth weight (BW) ≤ 1500 g with a video recording for GMA at 3–5 months CA were included. Poor-quality videos and infants with chromosomal and major congenital anomalies were excluded; definitions for these anomalies were adapted from the Canadian Neonatal Network Abstractor Manual [25] and generally included involvement of a significant part of a body system, aneuploidy, or other known or suspected underlying genetic disorder. Infants with head ultrasound findings consistent with prematurity (e.g., periventricular leukomalacia or intraventricular hemorrhage) were included in the study. Baseline neonatal, sociodemographic, and follow-up data were collected for all participants.

### 2.4. General Movements Assessment

Infants were assessed between 3 and 5 months corrected age using the general movements assessment. The results of the clinical GMA were dichotomized into “normal” and “abnormal” in order to compare the clinical assessment with the automated movement analysis. In the retrospective cohort, the Hadders-Algra method [26] was used as the personnel available at the time were trained using this method. Using the Hadders-Algra method and a previous publication from the center assessing its predictive ability for motor impairment [27], infants found to have “definitely abnormal” movements were dichotomized to having an “abnormal” result, while those with “mildly abnormal”, “normal suboptimal”, and “optimal” movements were categorized as “normal”. In the prospective cohort, the Prechtl method [7] was used because the Pediatric physiotherapist had acquired advanced training in the method at the initiation of the study. Infants with absent fidgety movements or abnormal movements were considered “abnormal”, while those with fidgety movements present were considered “normal”.

### 2.5. Automated Movement Analysis

Two-dimensional videos were recorded using a digital camera in the neonatal follow-up clinic using an examination table. A standardized video acquisition protocol was used for the prospective cohort.

The video analysis consisted of five steps—screening, pre-processing to enable automated analysis, large displacement optical flow (LDOF) [28] to enable pixel tracking, using a skin model to segment the video, and extracting movement features. All videos were screened for quality by two independent reviewers (KR, SO). A standardized protocol was used for screening. In addition, the computer-based analysis, including the use of LDOF to track movements, has been described previously [29]. Velocities were calculated from the displacement of all skin pixels between consecutive frames of motion [28,30]. Extraction of movement indices was performed using automated video-based analysis at BRI. A total of 289 variables were extracted. A conference article describing the methodology has been published previously [29]. Key variables included quantity of motion (determined by the number of skin pixels that are moving in any frame as a percentage of the total number of skin pixels in that frame), distance, speed, and acceleration of the skin pixels from frame to frame in every direction, variability of motion, determined by the standard deviation of the quantity of motion. It is important to note that some variables were derived from 3D models and applied to a 2D system likely reduced their sensitivity in detecting motion (e.g., jerk index [31]).

Authors involved in the development and application of the movement analysis program were blinded to the outcomes of the study participants and outcome assessors were blinded to the automated analysis results.

### 2.6. Outcome

The primary outcome for this predictive model was a clinical diagnosis of CP, which was made by a Developmental Pediatrician using a combination of history, physical examination, and brain imaging where available [32].

In order to address potential selection bias from differential loss to follow-up, several measures were implemented, including multiple phone reminders to attend scheduled appointments, offering in-home visits to complete assessments, and automatic rescheduling of appointments in the event of a missed appointment. To minimize information bias, those performing automated analysis were blinded to the clinical outcomes of the participants. In addition, therapists performing the clinical GMA were blinded to the assessments performed at 18–24 months. Diagnoses of CP were all made by one of the two Developmental Pediatricians in the neonatal follow-up clinic who were not blinded to the clinical course as funding and resources were not available for dedicated assessments for the study. Data collection was performed by a single author (KR) and standardized data collection forms with pre-specified definitions were used to minimize uncertainty in information. Two sources of information, the electronic chart and neonatal follow-up clinic paper chart, were used for each patient to cross-check information and disagreements were settled by discussion with one of the Principal Investigators (PC)s. Other characteristics of CP, such as the Gross Motor Functional Classification Scale (GMFCS) and type of CP were also collected.

### 2.7. Statistical Analysis

#### 2.7.1. Baseline and Follow-Up Characteristics

Baseline and follow-up characteristics of the overall population were analyzed using descriptive statistics. Student’s *t*-test and Wilcoxon Rank Sum test were used for continuous variables and the mean and standard deviation or median and inter-quartile range for normally and non-normally distributed data, respectively, were determined. For categorical variables, chi square test or Fisher exact test were used, as appropriate, and results presented as counts and percentages. Complete case analysis was used for missing data as it was assumed that missing data occurred at random. Missing data are reported in Section 3.

#### 2.7.2. Model Development

Logistic regression analysis was used to correlate movement variables with the outcome. Initially, unadjusted bivariate analysis was used and variables with *p* < 0.20 were retained for further selection in the multivariable regression model. Multicollinearity of retained variables was assessed using the variance inflation factor (VIF). Multicollinearity was present if VIF > 4.0. In such cases, only one member of a correlated set of variables was retained for the final model. Backward selection was then used to eliminate variables that did not meet a significance level of 0.20 to remain in the model [33,34]. Where more than one video was taken for a single patient, an average of the automated parameters was used for the analysis.

Bootstrapping taking 1000 samples of size 126 (the size of the original sample) with replacement was used as a means of internal validation. Harrell’s Optimism [35] was calculated from the bootstrapped samples and subtracted from the apparent c-statistic of the multivariable logistic regression model. A c-statistic change of ≤5% was considered acceptable.

Model fit was assessed using (1) the Hosmer–Lemeshow test of fit (*p* > 0.05 indicating acceptable fit) and (2) the C-statistic (C > 0.7 was considered acceptable discrimination) [36]. A receiver operator characteristic (ROC) curve was also constructed. In order to ensure that there was no overfitting, one variable was used for every 5–10 events, as previously described in the literature [37]. Influential observations were assessed using casewise diagnostics.

The probability level at which sensitivity and specificity were maximized was determined from the ROC curve. At this point, sensitivity, specificity, positive predictive value (PPV), negative predictive value (NPV), and accuracy were calculated for both the automated analysis and the clinical GMA. For the purposes of comparison, only the prospective cohort using the Prechtl method for clinical GMA was used [26]. One pediatric physiotherapist with >10 years of experience in recording and interpreting the GMA was involved in scoring the videos. At the time of the clinical GMA, the assessor was not blinded to the clinical course of the participants in the NICU. Between 3 and 5 months CA, the Prechtl scoring algorithm consists of three classifications—normal, abnormal, and absent fidgety movements—based on the quality of general movements was used [38]. For this study, children scored as “absent fidgety movements” were considered abnormal as this is most predictive of CP and there were no children in our cohort with “abnormal” fidgety movements.

All statistical analyses were conducted using SAS Studio version 3.4 (SAS Institute, Cary, NC, USA).

### 2.8. Sample Size Estimation

Based on the primary outcome rate of CP in the retrospective study of 22/152 (14%), in order to use 3–4 movement variables that capture the three broad concepts of GM—complexity, fluency, and variability [39]—5–10 cases are required per movement variable. Therefore, the number of patients with CP would have to be 15–40. Based on the rate of CP from our retrospective study, the total number of patients required for the prospective study was 107–285. Assuming a 10% loss to follow-up based on previous data [8], the number of patients required for recruitment was 118–317 patients.

## 3. Results

Figure 1 shows the flow of study participants. One hundred and seventy-nine participants were approached for consent and 148 were recruited for the study. Of those recruited for the study, 19 were lost to follow-up (12.8%) and they were excluded from the study sample. Overall, 100 participants were included prospectively. From the retrospective database, 152 participants were eligible for the study. Eight participants had two videos.

Table 1 shows the baseline neonatal and follow-up characteristics of the cohort. Overall, the median GA was 27^4/7^ weeks (range 25^6/7^–29^2/7^ weeks) and birth weight was 960 g (range 769–1215 g). Abnormal GMA was noted in 41 of 252 participants (16.3%).

The median CA at the follow-up visit was 18.6 months (range 18.2–19.7 months). The overall rate of CP was 11.5% (29/252). Most children with CP had spastic hemiplegia and approximately half of the retrospective cohort were ambulatory. There were several missing GMFCS values for the prospective cohort because of the inability to assign GMFCS before the age of 2 when the study ended [40]. For those who had GMFCS available, these were assigned after a 2-year visit.

After performing the variable selection, four variables appeared to provide the best model for CP prediction—the mean velocity in the vertical direction (V_my_), median quantity of motion (Q_med_), standard deviation of the quantity of motion (Q_sd_), and the minimum quantity of motion (Q_min_)—and were also independently associated with the outcome (Table 2).

Figure 2 shows the ROC curve for the multivariable logistic regression model using the four variables. The C-statistic was 0.74 and met the a priori cut-off, indicating a good fit. The Hosmer–Lemeshow test of fit also indicated good fit (χ^2^ 10.4, df = 8, *p* = 0.24). Casewise diagnostics revealed no influential outliers. Internal validation using bootstrapping revealed a Harrell’s Optimism of 0.04 and therefore a corrected C-statistic of 0.71 (5.6% change). Thus, the model was deemed internally valid.

Table 3 shows the predictive value of the automated GMA for CP. For comparison, the predictive value for CP using the Prechtl method of clinical GMA is shown in the same table.

## 4. Discussion

In this ambispective study, we developed and assessed a novel automated movement analysis model for predicting CP using four movement variables. Our automated analysis is able to predict CP with a sensitivity that is lower than the clinical GMA. However, there is a wide margin of error in the sensitivity of the clinical GMA, and this relates to experience with GMA (REF). In order to generalize implementation of the GMA, an automated tool may offer more precision in measurement compared to the human perceived gestalt perception that the clinical GMA relies on. Identification of ways to enhance the generalizability of the gold standard GMA is key, allowing for earlier identification of CP and our data will contribute to a growing body of literature on this topic.

The movement variables that appeared to have the greatest correlation with CP described the quantity of motion of the infant (Q_med_, Q_min_) and the variability of that motion (Q_sd_). This is in keeping with our hypothesis and with previous studies. For example, Adde et al. [9] first described the correlation of the reduced quantity of motion and reduced variability of this quantity of motion with the absence of fidgety movements. This was followed by similar findings in predicting CP early, but interestingly it was a combination of these two variables that provided improved predictive performance [41]. A higher Q_sd_ would indicate more variability in larger movements of the limbs, while a lower Q_sd_ would indicate less variability in larger movements and more subtle, lower amplitude midline movements, such as those seen in the fidgety period [7,38,42]. In the largest study of automated GMA to date by Ihlen et al. [15], a multivariable model using similar movement variables was found to perform with comparable sensitivity and specificity to the clinical GMA and cranial ultrasound. This study also demonstrated that the particular aspects of fidgety movements detected may be different from the clinical assessment; although, both had a high rate of false positives. Given that our study also showed a high false-positive rate, it is likely that automated analysis inherently detects different movements from the clinical assessment. As well, similar to this study, a cluster of variables was found to function better than a single variable to predict CP [15]. One variable that has not seemed to predict CP in our study is the variability in the centroid of motion, which in one study group appeared to independently predict both GMA and CP [43]. It is possible that this was because of a technical difference (e.g., skin silhouette was different, background conditions were different). It is also possible that other variables in our study were simple stronger predictors. In addition, higher velocity (V_my_) is associated with a higher risk of CP. Although this was not hypothesized, this is also seen clinically where the transition to fidgety movements is characterized by slower, lower amplitude, and more complex movements [7]. Thus, our model may capture the less fluent characteristics of abnormal fidgety movements. In contrast, limb velocities were found to be lower in infants with CP in one study [10]. However, this study was conducted using 3D motion capture with detectors placed on the infants’ limbs rather than computer vision. The placement of items on the infant’s skin may affect their movement. It is possible that by using limb-specific automated analysis, more specific GMA-related movements would be detected [44].

As a result of the relatively low prevalence of CP in our preterm population, the PPV of the automated and clinical assessments was low while the NPV was high. However, the PPV of the clinical GMA remains significantly higher than the automated movement analysis, which is a reflection of the lower sensitivity of this technology currently. Perhaps with the introduction of improved technology, such as skeleton models and 3D video capture, sensitivity can be increased [45,46,47]. The low prevalence may also relate to the decreased rate or plateau in CP rates in the preterm population [48,49,50], which is likely to result in screening tools having higher NPV than PPV. However, one study does report that while moderate to severe CP rates have likely decreased since the introduction of interventions such as antenatal magnesium sulfate, mild cases of CP may, in fact, be higher [51]. This is a limitation in our study as data were collected at 2 years and mild CP may not yet be detected.

This model has potential applications in the neonatal follow-up clinic. Specifically, smartphone applications are now available to record infant movements and upload them to a central database [52,53,54]. Thus, automated movement analysis could be used to pre-screen infants prior to a clinic visit and identify children who may need more intensive follow-up for motor impairment. The cost of and access to automated movement analysis technology may once have been somewhat prohibitive in terms of use on a larger scale. With new technology ever evolving, and the capacity to automate with simple and basic equipment, such as a digital camera and standard computer, the use of automated movement analysis offers a powerful option for GMA, particularly where expertise in GMA is sparse. With the recent interest in technologies like this, it is even possible that at some point, technology like this could be used to monitor other aspects of neurodevelopment as well [55].

### Study Limitations

There are several study limitations. First, selection bias could have been introduced by the exclusion of those who refused to consent; as data for these infants are not available, it is unclear whether the refusal was systematic. Loss to follow-up was also seen prospectively in 12.8% of recruited participants, despite measures in place to retain as many participants as possible. If a child was lost to follow-up, their data were excluded from the study. Based on previous studies, loss to follow-up occurs more with families who have limited resources, support, and capacity [56]. These characteristics also contribute to a greater degree of compromise with neurodevelopmental outcomes, and therefore bias could be introduced. However, cognitive and behavioral outcomes seem to be more affected than motor outcomes, minimizing the effect of this loss to follow-up on our study results [57].

Secondly, as the diagnosis of CP was given before the age of two, there may be instances where milder cases of CP were not identified. In addition, the GMFCS level is assigned typically after the age of two and therefore a lot of these data were not available [58]. Since this study utilized a closed design and no data were recorded after December 2019, it is possible that additional diagnoses of CP and a better understanding of function could have been obtained after the end of the study.

In the retrospective portion of the study, the rate of CP was higher than in the prospective study (14% versus 7%, respectively). Clinical studies have shown variability in the rates of CP [59] and findings in this study may be a reflection of this variability. On the other hand, it is possible that this may be reflective of decreased brain injury in preterm infants [48,49,50]. Inconsistencies between the two time periods may limit the generalizability of this study. In addition, there were very few cases of “abnormal” GMA in the prospective cohort and different clinical GMA tools (i.e., Hadders-Algra versus Prechtl) used in each cohort. Therefore, it was not possible to compare specific movement parameters in the normal GMA group and the abnormal GMA group. This analysis has been performed before for the quantity of motion in other studies, however, and correlations in this parameter with clinical GMA have been shown [9,22].

Lastly, these data do not capture non-CP motor impairments that have been seen more frequently in our preterm populations [48]. Conditions such as developmental coordination disorder were not assessed in this study. Future studies looking at all motor impairments would likely be more helpful in a neonatal follow-up setting.

## 5. Conclusions

This study shows that a simple, novel automated movement analysis may be able to predict CP using four kinematic features describing the overall quantity of motion, the variability of that motion, and its speed. Future studies will focus on further validation in the preterm population as well as validation in term infants with conditions that predispose them to motor impairment, such as hypoxic-ischemic encephalopathy.

## Figures and Tables

**Figure 1 children-09-00843-f001:**
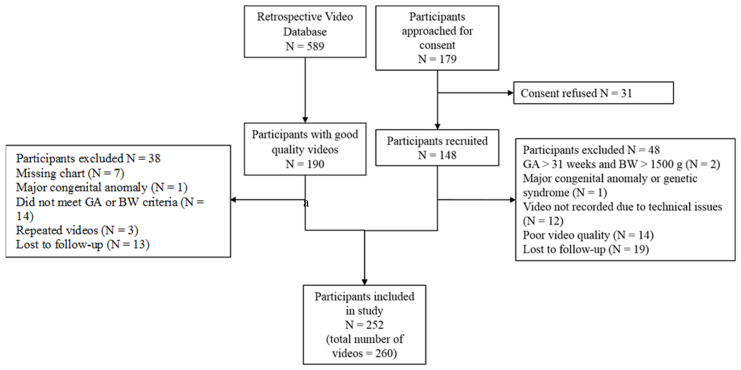
Flow diagram of study population. These videos were included in the analysis.

**Figure 2 children-09-00843-f002:**
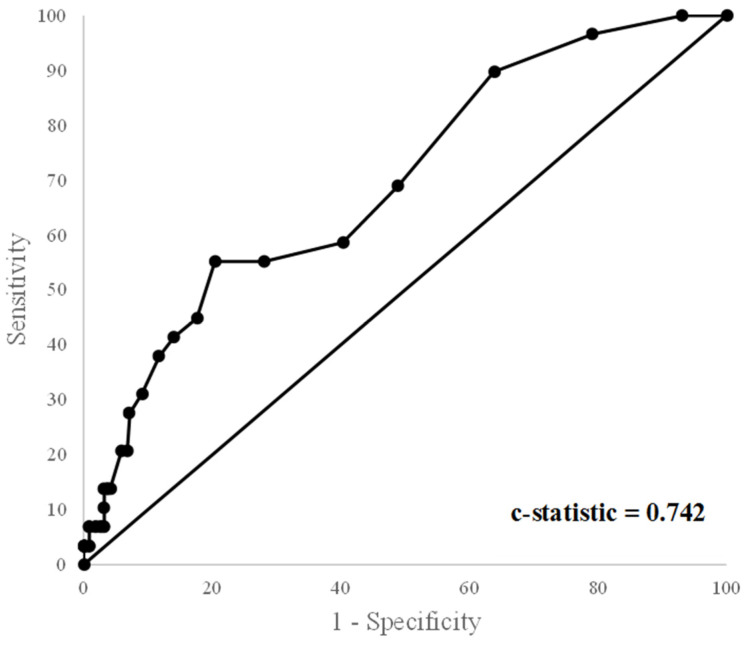
Receiver operator characteristic Curve for multivariable model.

**Table 1 children-09-00843-t001:** Baseline and follow-up characteristics of study population.

Characteristic	Overall (N = 252)	Missing, N (%)	Retrospective Study (N = 152)	Prospective Study (N = 100)
GA (weeks), median (IQR)	27.50 (25.86, 29.29)	0	27.7 (26, 29)	27.4 (25.2, 29.6)
BW (grams), median (IQR)	960 (769, 1215)	0	955 (764, 1235)	997 (737, 1287)
SGA, N (%)	20 (7.93)	0	12 (7.89)	8 (8)
Male sex, N (%)	125 (49.60)	0	79 (52)	46 (46)
BPD at 36 days, N (%)	35 (13.89)	6 (2.38)	25 (16)	10 (10)
Postnatal systemic steroids, N (%)	18 (7.14)	2 (0.79)	9 (5.9)	9 (9)
Home oxygen, N (%)	22 (8.73)	6 (2.38)	16 (10)	6 (6)
IVH ≥ grade III, N (%)	33 (13.10)	0	25 (16)	8 (8)
PVL, N (%)	7 (2.78)	0	5 (3.2)	2 (2)
Hydrocephalus requiring drainage, N (%)	17 (6.75)	0	12 (7.9)	5 (5)
ROP > stage III, N (%)	10 (3.97)	2 (0.79)	7 (4.61)	3 (3)
NEC ≥ stage II/III, N (%)	12 (4.76)	1 (0.4)	6 (4)	6 (6)
Meningitis, N (%)	10 (3.97)	1 (0.4)	4 (2.63)	6 (6)
PDA, N (%)	56 (22.22)	1 (0.4)	37 (24.34)	19 (19)
CA at GMA, median (IQR)	3.73 (3.40, 4.03)	0	3.81 (3.41, 4.21)	3.58 (3.28, 3.97)
Abnormal GMA ^a^, N (%)	41 (16.27)	8 (3.17)	32 (21)	9 (9)
CA at 18-month visit, median (IQR)	18.63 (18.2, 19.67)	13 (5.16)	18.5 (18.2, 19.4)	18.82 (18.27, 20.00)
BSID-III Motor composite score, median (IQR)	94 (85, 100)	13 (5.16)	97 (88, 100)	91 (82, 97)
BSID-III Motor composite score < 85, N (%)	44 (17.46)	12 (4.76)	22 (14.47)	22 (22)
BSID-III Motor composite score < 70, N (%)	19 (7.54)	12 (4.76)	12 (7.89)	7 (7)
Diagnosis of CP, N (%)	29 (11.5)	2 (0.79)	22 (14)	7 (7)
Type of CP, N (%)		3 (10.34)		
Spastic hemiplegia	14 (48.27)		10 (43)	4 (57)
Spastic diplegia	5 (17.24)		5 (22)	0 (0)
Spastic quadriplegia	2 (6.90)		2 (8.7)	0 (0)
Dystonic	2 (6.90)		2 (8.7)	0 (0)
Mixed	2 (6.90)		1 (4.3)	1 (14.3)
Other	1 (3.45)		2 (9)	1 (14.3)
GMFCS, N (%)		8 (27.59)		
I–II	10 (34.48)		10 (45)	0 (0) ^b^
III–V	11 (37.93)		10 (45)	1 (14.3)

BSID-III = Bayley Scales of Infant Development, 3rd edition; BPD = bronchopulmonary dysplasia; BW = birthweight; CA = corrected age; CP = cerebral palsy; GA = gestational age; GMA = general movements assessment; GMFCS = Gross Motor Function Classification Scale; IVH = intraventricular hemorrhage; NEC = necrotizing enterocolitis; PDA = patent ductus arteriosus; PVL = periventricular leukomalacia; ROP = retinopathy of prematurity; SGA = small for gestational age. ^a^ The Hadders-Algra method of GMA was used in the retrospective portion of the study, while the Prechtl method of GMA was used in the prospective portion of the study. The same assessor administered the GMA for both portions of the study. ^b^ GMFCS may not have been assigned as yet due to age of participants at study end date.

**Table 2 children-09-00843-t002:** Univariate and multivariate logistic regression models.

Predictor Variable	β Coefficient	CP OR (95% CI)	Test Statistic	*p*-Value
**Univariate Logistic Regression**				
V_my_	0.78	2.18 (1.19, 3.98)	6.36	**0.012**
Q_med_	0.07	1.07 (0.99, 1.16)	3.13	0.08
Q_sd_	1.54	4.66 (1.84, 11.80)	10.52	**0.0012**
Q_min_	−1.079	0.34 (0.14, 0.81)	6.01	**0.014**
**Multivariable Logistic Regression**				
Omnibus likelihood ratio (χ^2^)			22.98	**0.0001**
V_my_	0.75	2.12 (1.15, 3.92)	5.75	**0.016**
Q_med_	0.08	1.08 (0.99, 1.17)	3.32	0.068
Q_sd_	1.13	2.12 (1.15, 3.92)	4.96	**0.026**
Q_min_	−0.89	0.41 (0.16, 1.03)	3.63	0.057

CI = confidence interval; CP = cerebral palsy; OR = odds ratio; Q_med_ = median quantity of motion; Q_min_ = minimum quantity of motion; Q_sd_ = standard deviation of quantity of motion; V_my_ = mean velocity of movement in the vertical direction. Bolded = *p* < 0.05.

**Table 3 children-09-00843-t003:** Predictive value of multivariable logistic regression model for cerebral palsy.

	Sensitivity, % (95% CI)	Specificity, % (95% CI)	PPV, % (95% CI)	NPV, % (95% CI)
Automated movement analysis	55.17 (35.69, 73.55)	79.64 (73.72, 84.74)	26.23 (18.95, 35.09)	93.12 (89.99, 95.32)
Clinical GMA ^a^	85.71 (42.13, 99.64)	96.43 (89.92, 99.26)	66.67 (38.73, 86.35)	98.78 (92.95, 99.80)

CI = confidence interval; GMA = general movements assessment; NPV = negative predictive value; PPV = positive predictive value; Q_med_ = median quantity of motion. ^a^ Predictive values are reflective of Prechtl method GMA used in prospective study cohort only.

## Data Availability

Data supporting reported results are available upon request and are housed on a secured server with the study’s authors K.R. and S.O.

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
