# Peer review of "Automated Movement Analysis to Predict Cerebral Palsy in Very Preterm Infants: An Ambispective Cohort Study"

_children, 2022, doi:10.3390/children9060843_

Round 1
Reviewer 1 Report
Thank you very much for inviting me to do this review.
GMA is one of the tools that is being used the most in terms of early detection of CP and its use is increasing in all countries of the world. Therefore, the topic of this article is very current. Even so, I have doubts in some aspects of each section
Abstract
- There should be no acronyms in this section.
- From my point of view, it is very risky to talk about the benefits in premature infants, when moderate and late preterms are not collected.
- In addition, that the data of his study show worse data than the face-to-face of GMA. So I think it should be used only in some cases.
Introduction
- At the end of my text, I leave you some more references on GMA and technology, for your use in the introduction and in the discussion.
- We should talk about the population groups of preterm infants that have been investigated and of all premature infants.
- Evaluate whether to change the title to adjust the population.
Methodology
- The ethics committee registration number is missing.
- The exclusion criteria should be more detailed, in relation to neurologic major alterations (leukomalacia, intracranial hemorrhage, ...).
- This section should be better organized, since it does not follow a logical sequence; in addition to creating a specific one on GMA.
- It should be specified in this section when it was classified in GMFCS (it is explained vaguely in the results and should be in this section; and the use of HINE at 9 months to make the approximation would have been interesting).
- It should be explained why using two methods of GM evaluation
Results
- Why are there subjects with two videos?
- It seems to me that very few subjects are excluded for major neurological reasons, when this type of population has a high risk, which, together with low weight, increases the neurological risk much more.
- There is talk of neurological conditions that have been studied in the MGs and that modify their results, but nothing has been said about them in the inclusion/exclusion criteria.
- The percentages of sensitivity and specificity that have been obtained are very low.
Discussion
- This section should be rewritten.
- It is very scarce and all the evidence on GMA and existing technology should be used.
Conclusion
- It's fine and it's what should appear in the abstract.
Reviewer 2 Report
Please see attached document

Reviewer 3 Report
The authors reported the usefulness of a novel 2D video-based analysis method as an initial screen for cerebral palsy in preterm infant. It's an interesting theme. To make it more meaningful and useful, I request to revise several points.
Materials and method
#1 Citation is difficult to access and detailed information on video-based analysis methods is difficult to access. Please change citation(Reference No.) 20 to the paper published in a peer-reviewed journal. If that is not possible, please describe more about the video-based analysis methods in this manuscript.
#2 It's unclear how to the statistical analysis including the eight participants who had two videos. In general, it is not appropriate to include multiple data from only some participants.
Results
#3 It is very interesting what the variables obtained from the video analysis reflect in traditional GMA. Although the authors have described it in the discussion, please investigate the direct relationship. I recommend the authors to investigate and compare the video variables data of normal and abnormal GMA group.
Round 2
Reviewer 1 Report
Great job!!! I think this manuscript is better than the previous one.
But I think you need to explain better that you used the Hadders-Algra's and Pretchl's methods to get a dichotomous score to differentiate babies with abnormal general movements.
I send you the list of references
Automated pose estimation captures key aspects of General Movements at 8-17 weeks from conventional videos
Deep learning‑based quantitative analyses of spontaneous movements and their association with early neurological development in preterm infants
Detection of Infantile Movement Disorders in Video Data Using Deformable Part-Based Model
General Movement Assessment from videos of computed 3D infant body models is equally effective compared to conventional RGB video rating
In-Motion-App for remote General Movement Assessment a multi-site observational study
Movement recognition technology as a method of assessing spontaneous general movements in high risk infants
Movidea A Software Package for Automatic Video Analysis of Movements in Infants at Risk for Neurodevelopmental Disorders
Spontaneous movements in the newborns a tool of quantitative video analysis of preterm babies
Supine lying center of pressure movement characteristics as a predictor of normal developmental stages in early infancy
Technology-assisted quantification of movement to predict infants at high risk of motor disability A systematic review
The future of General Movement Assessment The role of computer vision and machine learning – A scoping review
Usability and inter-rater reliability of the NeuroMotion app A tool in General Movements Assessments
Analysis of motor development within the first year of life 3-D motion tracking withoutmarkers for early detection of developmental disorders
Characteristics of general movements in preterm infants assessed by computer-based video analysis
Regards
Reviewer 3 Report
Point 1; I have confirmed that the manuscript is properly revised.
Point2; I can understand repeat videos were taken when there was uncertainty around the movements seen in the first video or if the first video was taken at a very early time point in the fidgety period. In these case, I think the data from the good quality video (not average) should be used for the statistical analysis. If the authors analyze using the average, it is acceptable. But the authors should describe in the method section that the analysis was performed using the average value in case of having multiple points data.
Point 3; I think this is a necessary investigation, when the authors propose this as an automated screening tool based on the concept of the GMA. But if it cannot be done, there is no help for it. Please add the reason why the authors cannot investigate it in the discussion or in the limitation section.
